# Task-recency bias strikes back: Adapting covariances in Exemplar-Free Class Incremental Learning

**Grzegorz Rypeść***
IDEAS NCBR
Warsaw University of Technology
`grzegorz.rypesc@ideas-ncbr.pl`

**Sebastian Cygert**
IDEAS NCBR
Gdańsk University of Technology
`sebastian.cygert@ideas-ncbr.pl`

**Tomasz Trzciński**
IDEAS NCBR
Warsaw University of Technology
Tooploox

**Bartłomiej Twardowski**
IDEAS NCBR
Autonomous University of Barcelona
Computer Vision Center

## Abstract

Exemplar-Free Class Incremental Learning (EFCIL) tackles the problem of training a model on a sequence of tasks without access to past data. Existing state-of-the-art methods represent classes as Gaussian distributions in the feature extractor's latent space, enabling Bayes classification or training the classifier by replaying pseudo features. However, we identify two critical issues that compromise their efficacy when the feature extractor is updated on incremental tasks. First, they do not consider that classes' covariance matrices change and must be adapted after each task. Second, they are susceptible to a task-recency bias caused by dimensionality collapse occurring during training. In this work, we propose *AdaGauss* – a novel method that adapts covariance matrices from task to task and mitigates the task-recency bias owing to the additional anti-collapse loss function. *AdaGauss* yields state-of-the-art results on popular EFCIL benchmarks and datasets when training from scratch or starting from a pre-trained backbone.

## 1 Introduction

Continual learning (CL), an essential area of machine learning, focuses on developing algorithms that can learn progressively from a continuous stream of data and adapt to new tasks while retaining previously acquired knowledge. This paradigm is paramount for creating systems capable of lifelong learning, much like humans, and robust in dynamic environments where data distribution evolves over time. A significant challenge within CL is exemplar-free class incremental learning (EFCIL) [41, 26], which requires the model to incorporate new classes without storing previous data samples (exemplars). This approach is especially relevant in scenarios with privacy constraints or limited storage capacity, as it compels the model to retain knowledge and prevent catastrophic forgetting [9, 27] solely through internal mechanisms, such as knowledge distillation [12, 21, 45, 51, 24], parameter regularization [17, 2, 6], expanding neural architecture [44, 53, 32, 3] or generative replay [40, 14, 28].

Recent state-of-the-art methods designed for EFCIL often represent classes as Gaussian distributions in the latent space of the feature extractor. That enables an inference using Bayes classifier [10, 35] or training a linear classifier using pseudo-prototypes sampled from these distributions [24, 31, 51, 37]. However, we present in this work that these methods have multiple shortcomings and can be improved. First, they assume that covariance matrices of past classes are constant across incremental training.

---

*Code: `https://github.com/grypesc/AdaGauss`

38th Conference on Neural Information Processing Systems (NeurIPS 2024).

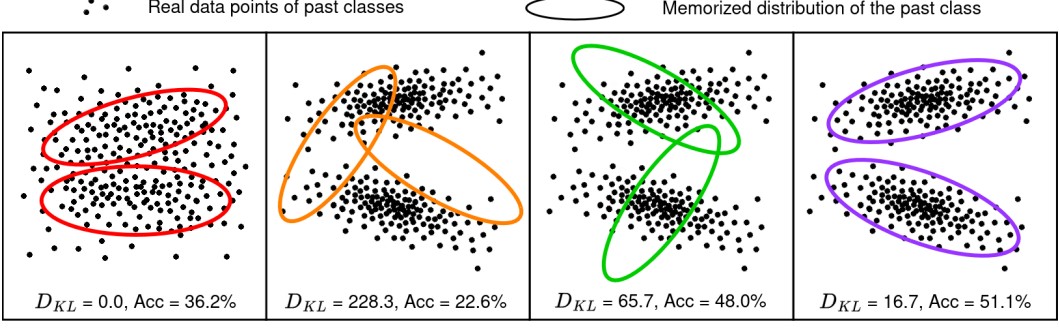

Figure 1: Latent space visualization, average accuracy after the last task, and symmetrical KL divergence between memorized and ground truth distributions for ResNet18 trained sequentially on ImagenetSubset dataset split into ten tasks. Freezing the feature extractor prevents changes in data distribution but results in inseparable classes. When the network is trained on incremental tasks (unfrozen), the ground truth distributions change and do not match the memorized ones. A suitable CL method should adapt the mean and covariance of distributions to retain valid decision boundaries.

However, as presented in Fig. 1, when the feature extractor is updated on incremental tasks (it is unfrozen), distributions of previous classes change and no longer match the memorized ones. Suitable methods must adapt both means and covariances. EFC [24] predicts drift (change) only of the distribution mean and points out that adapting covariances is an open question. Second, the methods suffer from a dimensionality collapse [30, 16], which is more significant in early tasks. That makes old classes' covariances to be of lower rank than those from recent tasks, which introduces errors while inverting the matrices for the classification, leading to increased task-recency bias. We explain this in detail in Sec. 3.2.

This work focuses on the challenging problems of adapting classes' covariances and overcoming dimensional collapse in EFCIL. We are the first to introduce a method that adapts the mean and covariance of memorized distributions, significantly reducing the error between memorized and ground truth distributions. We also overcome the dimensionality collapse of feature representations by introducing a novel anti-collapse loss, which alleviates the problem of task-recency bias. We dub the resulting method *AdaGauss* - Adapting Gaussians. Our contributions are as follows:

- We analyze dimensionality collapse in EFCIL settings and explain that it leads to task-recency bias. We introduce a novel anti-collapse loss to prevent it.

- We show that knowledge distillation techniques in EFCIL provide different representation strengths of the feature extractor. We are the first to utilize knowledge distillation through a learnable projector network in EFCIL.

- Based on these findings, we propose *AdaGauss*, a novel method to adapt both means and covariances of memorized class distributions, which results in state-of-the-art results when the model is trained from scratch or starting from a pre-trained weights.

## 2 Related works

**Semantic drift.** We investigate offline, EFCIL setting [26] focusing on keeping the network size constant, where no task information is available at test time. Regularization-based approaches penalize changes to important neural network parameters [17, 6, 47, 22] or use distillation techniques to regularize neuron activations [21, 45, 51, 24]. However, even with knowledge distillation, the features from old classes will change, causing catastrophic forgetting [9, 27]. Therefore, few works tried to predict these changes by approximating their semantic drift [45, 24, 15, 36]. However, those strategies' limitations are that they adapt only prototypes, ignoring changes in covariance matrices, which we experimentally show is suboptimal. As predicting the drift is challenging, many methods focus on scenarios where the backbone is frozen after the first task [4, 31, 23, 29, 10]. However, this prevents the feature extractor to adapt to new tasks [24]. We show that it is possible to change the feature extractor and adapt the covariance matrices of classes.

**Task-recency bias.** Another challenge in CL is a task-recency bias, where the model is biased towards classifying classes from new tasks [13, 26, 50]. While some works approached this problem using exemplars [1, 43, 13, 48] the problem is amplified in an exemplar-free setting. Some works considered prototype replay, which maintains the decision boundary between classes [31, 51, 37, 39, 38, 53]. To improve this strategy, PASS [52] included prototype augmentation, and EFC [24] updates their prototypes after each task. In this work, we point out that the cause for task-recency bias in the EFCIL scenario is the dimensionality collapse of the feature extractor, leading to numerical instabilities when inverting covariance matrices.

**Dimensionality collapse.** Recent works revealed that supervised learning exhibits signs of neural collapse [30, 16], where a large fraction of features' variance is described only by a small fraction of their dimensions. Since then, several studies [5, 7, 46, 16] showed that utilizing additional MLP projector is a crucial component to alleviate the collapse of the representations and improve their transferability. Another implication of the neural collapse in CL is that it becomes challenging to invert covariance matrices. Existing methods add a constant value to the diagonal [35, 24, 51] of the covariance matrices or utilize shrinking [10] to prevent that. On the contrary, we propose an anti-collapse loss, which is more elegant and does not artificially alter covariance matrices.

## 3 Method

### 3.1 Exemplar-Free Class-Incremental Learning (EFCIL)

Class-Incremental Learning (CIL) scenario considers a dataset split into $T$ tasks, each corresponding to the non-overlapping set of classes $C_1 \cup C_2 \cup \cdots \cup C_T = C$ such that $C_t \cap C_s = \emptyset$ for $t \neq s$. In Exemplar-Free CIL (EFCIL), during a training step $t$, we have only access to current task data $D_t = \{(x, y) | y \in C_t\}$ and we cannot store any exemplars from the previous steps. The objective is to train a model that discriminates between old ($< t$) and new classes combined. We assume a task-agnostic evaluation [41, 26], where the method does not know the task id during the inference.

### 3.2 The three observations that motivate towards *AdaGauss*

In this section, we provide an insight into problems with current EFCIL methods. We train the standard ResNet18 [11] network on the ImagenetSubset dataset divided into ten equal tasks. We point out that: **1.** covariance of class distributions during CL sessions change and must be adapted; **2.** the task recency bias comes from the differences in representational strength of the model; **3.** when training from scratch in EFCIL, the models are susceptible to dimensionality collapse.

**Observation 1**. As illustrated in Fig. 1, training the feature extractor on incremental tasks makes memorized distribution not match the ground truth (GT) ones. More specifically, the mean and covariance of GT change, and to keep valid decision boundaries, both memorized means and covariances must be adapted. That decreases symmetrical KL divergence between memorized and GT distributions, thus increasing average accuracy after the last task. However, existing state-of-the-art methods [36, 24, 51, 52] do not adapt covariance matrices, while others [31, 10, 55, 54] freeze the feature extractor after the initial task, which does not guarantee separability of classes from new tasks (first image in Fig. 1).

**Observation 2**. When training the feature extractor with different knowledge distillation methods (feature [53, 52, 45], logit [21, 33], projected [18]), representational strength of the feature extractor increases with each task, as presented in Fig. 2. That makes memorized covariance matrices of late tasks have a higher rank than those from early tasks, as presented in Fig. 3. When these matrices are inverted, the opposite happens - due to numerical instabilities, norms of inverted covariance matrices of early tasks will be greater. That causes task-recency bias as presented in Fig. 4. In the case of Bayes classification [35, 10], the Mahalanobis distance is much higher for early tasks, whereas in the case of sampling pseudo-prototypes [24, 51] the logits for recent tasks are higher, what skews classification towards recent tasks. This bias differs from already well-studied linear head bias [13, 43, 48], as it occurs at the level of the representations, where no linear head and no exemplars are utilized.

**Observation 3**. Fig. 3 also presents that feature extractor suffers from dimensionality collapse [30, 16] as ranks of covariance matrices are much lower than the latent space size (512 for ResNet18). That makes classes covariance matrices non-invertible. That, in turn, disallows the calculation of Mahalanobis distance, likelihood, and sampling from such collapsed distribution. In order

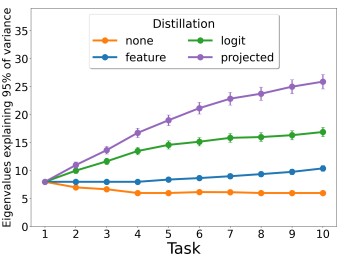 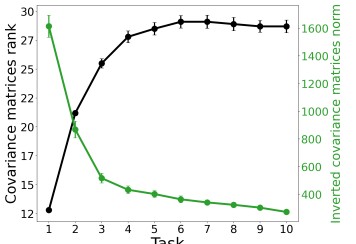 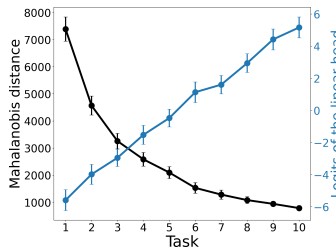

Figure 2: The representational strength of ResNet18 trained on 10 tasks of ImagenetSubset dataset split into 10 tasks for different knowledge distillation methods. After each task, we measure how many eigenvalues sum to 95% variance of all features provided.

Figure 3: Average rank of memorized covariance matrices of classes after each task (black) on ImagenetSubset for logit distillation. Norm of these matrices when inverted (green). Lower rank leads to larger values in inverses of covariance matrices due to numerical instabilities.

Figure 4: Average Mahalanobis distance between memorized distributions and joint dataset per each task after the last task (black) and average logit value on linear head trained by sampling prototypes from memorized distributions. There is a visible task-recency bias.

to overcome this issue, the existing methods utilize shrinking [10] or add a constant value to the diagonal [35, 24, 51] of the covariance matrices to prevent that. However, these techniques artificially alter classes' distributions, introducing additional hyperparameters and a new source of errors accumulating during long CIL sessions. A more elegant solution would directly prevent the dimensionality collapse of the feature extractor during training while preserving the class separability provided by cross-entropy.

### 3.3 *AdaGauss* method

Motivated by these three observations, we made the following decisions about *AdaGauss*. Based on the first observation, after training the feature extractor $F$ on an incremental task, we train an auxiliary network (adapter), which we utilize to adapt the means and covariances of old classes to the latent space of the new feature extractor. To perform knowledge distillation and improve the representation strength of the feature extractor (second observation), we utilize feature distillation through a learnable projector. In order to overcome the dimensionality collapse and task-recency bias, showcased by the second and third observations, we utilize a novel anti-collapse loss that regularizes the features' covariance matrix and prevents dimensional collapse. *AdaGauss* memorizes each class as a mean and covariance and performs Bayes classification as in [10, 35]. We provide a pseudo-code of our method in Alg. 1. Below, we explain the motivation and details of the method.

#### 3.3.1 Feature distillation through a learnable projector

Inspired by representational-learning [18], we utilize a feature distillation through a learnable projector to mitigate forgetting, which we refer to as projected distillation. As presented in Fig. 2, this distillation technique provides representations with a better eigenvalues distribution, thus decreasing the problem of task-recency bias compared to standard logit [21, 33] and feature [45, 53, 52] distillation techniques. As the projector, we utilize a 2-layer MLP network $\phi^{t \to t-1} : \mathbb{R}^S \to \mathbb{R}^S$ with hidden size $d$ times bigger than the latent space $S$. Following existing continual learning works [21, 45, 33], when training $F_t$ on minibatch $B$, we freeze $F_{t-1}$ trained on the previous task. Finally, we calculate our knowledge distillation loss as follows:

$$L_{PKD} = \sum_{i \in B} ||\phi^{t \to t-1} \left( F_t \left( x_i \right) \right) - F_{t-1} \left( x_i \right) ||^2. \tag{1}$$

#### 3.3.2 Overcoming dimensionality collapse

As described in Sec. 3.2, existing methods for EFCIL that represent classes as Gaussian distributions suffer from dimensionality collapse, which leads to task-recency bias caused by the fact that ranks of covariance matrices are different amongst the tasks. To overcome the collapse, we encourage the

feature extractor to produce features whose dimensions are linearly independent. Therefore, in each task, we directly optimize the covariance matrices of features produced by $F_t$ to be positive-definitive by the diagonal of the Cholesky decomposition of covariance of each training minibatch to be positive. More precisely, let $S$ be the size of the feature vectors and $a_i$ be $i$-th element of the diagonal of a Cholesky decomposition of minibatch's covariance matrix. We formulate the anti-collapse loss $L_{AC}$ in the form:

$$L_{AC} = -\frac{1}{S} \sum_{i=1}^{S} \min(a_i, 1) \tag{2}$$

This loss forces Cholesky's decomposition of covariance of each minibatch to have diagonal entries greater than 1. Therefore, they are positive, and the covariance matrix is positive-definite due to the property of Cholesky decomposition. More on the definition of $L_{AC}$ in Appendix, Sec. A.2).

### 3.3.3 Training the feature extractor

In each task $t$, we train all parameters of the feature extractor $F_t$ together with additional projector $\phi$ used for knowledge distillation. Following most works [10, 35, 51, 31, 21], we utilize popular cross-entropy loss $L_{CE}$ to discriminate between classes. The final loss function is:

$$L = L_{CE} + L_{AC} + \lambda L_{PKD}, \tag{3}$$

where $\lambda \in \mathcal{R}$ is a plasticity-stability trade-off hyperparameter, similar to [21].

After training the feature extractor, we represent classes $C_t$ as multivariate Gaussian distributions in the latent space. More precisely, we represent any class $c \in C_t$ as $\mathcal{N}(\mu_c, \Sigma_c)$.

### 3.3.4 Adapting Gaussian distributions

After training of $F_t$ is completed, representations of old classes drifted [45] (changed) and no longer match memorized Gaussians. Therefore, we update memorized Gaussians representing past classes to recover ground truth representations. To do that, we train an auxiliary adaptation network $\psi^{t-1 \to t} : \mathbb{R}^S \to \mathbb{R}^S$ (called adapter), which maps features from the old latent space to the new one. We use only the current data from task $t$ for that. Training loss is:

$$L_\psi = \sum_{i \in B} ||\psi^{t-1 \to t}(F_{t-1}(x_i)) - F_t(x_i)||^2 + L_{AC}. \tag{4}$$

$L_{AC}$ is the same anti-collapse loss as used during the training of the feature extractor. After training the adapter, for each old class $c$, we sample from $\mathcal{N}(\mu_c, \Sigma_c)$ a set of $N$ points: $n_1, n_2, \ldots, n_N$, where $N \gg |S|$ and transform them through $\psi$ obtaining new set: $\{\psi(n_1), \psi(n_2), \ldots, \psi(n_N)\}$. We calculate adopted mean $\mu_c^{new}$ and covariance $\Sigma_c^{new}$ using new sets of data and update the old distribution as follows: $(\mu_c, \Sigma_c) = (\mu_c^{new}, \Sigma_c^{new})$ A pseudocode of the full *AdaGauss* method is presented in Alg. 1.

## 4 Experiments

**Datasets and metrics.** We evaluate our method on several well-established benchmark datasets. CIFAR100 [19] consists of 50k training and 10k testing images in resolution 32x32. TinyImageNet [20], a subset of ImageNet [8], has 100k training and 10k testing images in 64x64 resolution. ImagenetSubset contains 100 classes from ImageNet (ILSVRC 2012) [34]. We split these datasets into 10 and 20 equal tasks. Thus, each task contains the same number of classes, a standard practice in EFCIL [21, 45, 35, 24]. We also evaluate our method on fine-grained datasets: CUB200 [42] represents $11,788$ images of bird species, and FGVCAircraft [25] dataset consists of $10,200$ images of planes. We split fine-grained datasets into 5, 10, and 20 tasks. As the evaluation metric, we utilize commonly used average accuracy $A_{last}$, which is the accuracy after the last task, and average incremental accuracy $A_{inc}$, which is the average of accuracies after each task [26, 24, 10].

**Baselines and hyperparameters.** We compare our method to multiple EFCIL baselines. Well-established ones, like EWC [17], LwF [21], PASS [52], IL2A [51], SSRE [53], and the most recent and strong EFCIL baselines: FeTrIL [31], FeCAM [10], DS-AL [54] and EFC [24]. For the baseline

---

**Algorithm 1** *AdaGauss*: Adapting Gaussians in EFCIL

---

1: **Initialize:** Training data $(D_1, D_2, \ldots, D_T)$, $F_1$ (feature extractor), $\lambda$, $N$
2: Train $F_1$ on $D_1$ using $L_{CE} + L_{AC}$
3: **for** $c \in C_1$ **do**
4:     Obtain set of features: $O = \{F_1(x) : x, c \in D_1\}$
5:     Store $\mu_c = \text{mean}(O)$ and $\Sigma_c = \text{covariance}(O)$
6: **end for**
7: **for** $t = 2, 3, 4, \ldots$ **do**
8:     Initialize $\phi^{t \to t-1}$ (distiller), $\psi^{t-1 \to t}$ (adapter)
9:     Train $F_t$ on $D_t$ using $L = L_{CE} + L_{AC} + \lambda L_{PKD}$
10:     **for** $c \in C_t$ **do**
11:         Obtain set of features: $O = \{F_t(x) : x, c \in D_t\}$
12:         Store $\mu_c = \text{mean}(O)$ and $\Sigma_c = \text{covariance}(O)$
13:     **end for**
14:     Train adapter $\psi^{t-1 \to t}$ on $D_t$ using $L_\psi + L_{AC}$
15:     **for** $c \in \cup_{i=1}^{t-1} C_i$ **do**
16:         Sample $n_1, n_2, \ldots, n_N$ from $\mathcal{N}(\mu_c, \Sigma_c)$
17:         Calculate $\mu_c^{new}, \Sigma_c^{new}$ of set $\{\psi^{t-1 \to t}(n_1), \psi^{t-1 \to t}(n_2), \ldots, \psi^{t-1 \to t}(n_N)\}$
18:         $\mu_c = \mu_c^{new}$; $\Sigma_c = \Sigma_c^{new}$
19:     **end for**
20: **end for**

---

results on CIFAR100, TinyImageNet, and ImagenetSubset, we take the results reported in [24], while for FeCAM, we run its original implementation. For fine-grained datasets (CUB200, FGVCAircrafts), we run implementations provided in FACIL [26] and PyCIL [49] frameworks (if provided) or from the authors' repositories. We set default hyperparameters proposed in the original works. We utilize random crops and horizontal flips as data augmentation.

**Implementation details and reproducibility.** We utilize standard ResNet18 [11] as a feature extractor $F$ for all methods. We train it from scratch on CIFAR100, TinyImagenetSubset, and ImagenetSubset, while for experiments on fine-grained datasets, we utilize weights pre-trained on ImageNet. We implement our method in FACIL[26] benchmark[2]. We set $\lambda = 10, N = 10000, d = 32$ and add a single linear bottleneck layer at the end of the $F$ with $S$ output dimensions, which define the latent space. When training from scratch, we set $S = 64$, while for fine-grained datasets, we decrease it to 32, as there are fewer examples per class. We use an SGD optimizer running for 200 epochs with a weight decay equal to $0.0005$. When training from scratch, we utilize a starting learning rate (lr) of $0.1$, decreased by ten times after 60, 120, and 180 epochs. We train the adapter using an SGD optimizer with weight decay of 0.0005, running for 100 epochs with a starting lr of $0.01$; we decrease it ten times after 45 and 90 epochs.

We utilize a single machine with an NVIDIA RTX4080 graphics card to run experiments. The time for execution of a single experiment varied depending on the dataset type, but it was at most ten hours. We attach details of utilized hyperparameters in scripts in the code repository. We report all results as the mean and variance of five runs.

### 4.1 Results

**Training from scratch.** We present the baseline results and *AdaGauss* method when training from scratch in Tab. 1. We consider $T = 10$ and $T = 20$ equal tasks. We can see an improvement over the most recent state-of-the-art method - EFC [24]. We improve its results by 3.7% and 6.8% points in terms of average accuracy on ImagenetSubset split into 10 and 20 tasks, respectively. This improvement is also consistent in terms of average incremental accuracy - 5.1% and 7.5% points and on the other datasets. This increase can be attributed to the fact that EFC does not adapt covariance matrices from task to task (just means), which, as we showed in Sec. 3.2, is required to improve the results. Older method - IL2A [51], which does not adapt their classes representations (means and covariance matrices) method at all, achieves much lower results than our approach - 23.4% and 25.1% points lower average accuracy on ImagenetSubset.

---

[2] The code is provided in the Supplementary Materials and will be published upon acceptance.

Table 1: Average incremental and last accuracy in EFCIL when training the feature extractor from scratch. The mean of 5 runs is reported. Full results are in Tab. 5. We denote the best results **in bold**.

| Method | CIFAR-100 | | | | TinyImageNet | | | | ImagenetSubset | | | |
|---|---|---|---|---|---|---|---|---|---|---|---|---|
| | $T$=10 | | $T$=20 | | $T$=10 | | $T$=20 | | $T$=10 | | $T$=20 | |
| | $A_{last}$ | $A_{inc}$ | $A_{last}$ | $A_{inc}$ | $A_{last}$ | $A_{inc}$ | $A_{last}$ | $A_{inc}$ | $A_{last}$ | $A_{inc}$ | $A_{last}$ | $A_{inc}$ |
| EWC [17] | 31.2 | 49.1 | 17.4 | 31.0 | 17.6 | 32.6 | 11.3 | 26.8 | 24.6 | 39.4 | 12.8 | 27.0 |
| LwF [21] | 32.8 | 53.9 | 17.4 | 38.4 | 26.1 | 45.1 | 15.0 | 32.9 | 37.7 | 56.4 | 18.6 | 40.2 |
| PASS [52] | 30.5 | 47.9 | 17.4 | 32.9 | 24.1 | 39.3 | 18.7 | 32.0 | 26.4 | 45.7 | 14.4 | 31.7 |
| IL2A [51] | 31.7 | 48.4 | 23.0 | 40.2 | 25.3 | 42.0 | 19.8 | 35.5 | 27.7 | 48.4 | 17.5 | 34.9 |
| SSRE [53] | 30.4 | 47.3 | 17.5 | 32.5 | 22.9 | 38.8 | 17.3 | 30.6 | 25.4 | 43.8 | 16.3 | 31.2 |
| FeTrIL [31] | 34.9 | 51.2 | 23.3 | 38.5 | 31.0 | 45.6 | 25.7 | 39.5 | 36.2 | 52.6 | 26.6 | 42.4 |
| FeCAM [10] | 32.4 | 48.3 | 20.6 | 34.1 | 30.8 | 44.5 | 25.2 | 38.3 | 38.7 | 54.8 | 29.0 | 44.6 |
| DS-AL [54] | 40.8 | 54.9 | 31.7 | 43.2 | 33.6 | 47.2 | 26.5 | 41.6 | 46.8 | 58.6 | 36.7 | 48.5 |
| EFC [24] | 43.6 | 58.6 | 32.2 | 47.3 | 34.1 | 48.0 | 28.7 | 42.1 | 47.4 | 59.9 | 35.8 | 49.9 |
| AdaGauss | **46.1** | **60.2** | **37.8** | **52.4** | **36.5** | **50.6** | **31.3** | **45.1** | **51.1** | **65.0** | **42.6** | **57.4** |

Table 2: Average incremental and last accuracy in EFCIL fine-grained scenarios when utilizing a pre-trained feature extractor. We report the mean of 5 runs, while variances are reported in Tab. 6.

| Method | CUB200 | | | | | | FGVCAircraft | | | | | |
|---|---|---|---|---|---|---|---|---|---|---|---|---|
| | $T$=5 | | $T$=10 | | $T$=20 | | $T$=5 | | $T$=10 | | $T$=20 | |
| | $A_{last}$ | $A_{inc}$ | $A_{last}$ | $A_{inc}$ | $A_{last}$ | $A_{inc}$ | $A_{last}$ | $A_{inc}$ | $A_{last}$ | $A_{inc}$ | $A_{last}$ | $A_{inc}$ |
| EWC [17] | 21.6 | 38.2 | 15.8 | 32.6 | 12.3 | 27.2 | 24.3 | 44.0 | 14.3 | 34.5 | 10.9 | 27.9 |
| LwF [21] | 44.3 | 57.7 | 30.4 | 46.1 | 19.4 | 34.7 | 39.0 | 55.2 | 28.0 | 46.5 | 14.7 | 30.5 |
| PASS [52] | 34.5 | 48.6 | 27.0 | 42.3 | 18.1 | 36.9 | 33.3 | 48.9 | 26.4 | 41.0 | 13.9 | 28.1 |
| IL2A [51] | 36.9 | 51.3 | 29.4 | 45.5 | 20.8 | 35.1 | 39.4 | 49.1 | 27.3 | 45.1 | 14.2 | 28.7 |
| FeTrIL [31] | 41.9 | 53.2 | 36.9 | 48.2 | 34.6 | 45.3 | 46.0 | 58.5 | 40.5 | 53.4 | 32.5 | 43.3 |
| FeCAM [10] | 43.5 | 56.0 | 40.2 | 54.9 | 36.2 | 48.9 | 45.3 | 58.0 | 41.4 | 55.2 | 34.0 | 46.0 |
| DS-AL [54] | 49.4 | 61.9 | 45.8 | 59.1 | 41.4 | 53.8 | 50.6 | 62.7 | 42.6 | 56.4 | 34.2 | 46.7 |
| EFC [24] | 58.3 | 68.9 | 51.0 | 63.3 | 46.1 | 59.3 | 50.1 | 63.2 | 43.1 | 57.6 | 28.1 | 48.2 |
| AdaGauss | **60.4** | **69.2** | **55.8** | **66.2** | **47.4** | **60.6** | **53.3** | **64.0** | **47.5** | **58.5** | **34.8** | **48.6** |

Methods such as FeTrIL [31], FeCAM [10] and DS-AL [54] overcome the problem of distribution drift by freezing the feature extractor on the first task. However, it cannot adapt well to the new incremental tasks, resulting in poor plasticity and worse results than *AdaGauss* and EFC [24]. FeTrIL achieves 14.9% and 16.0% points lower average accuracy on ImagenetSubset, while FeCAM - 12.4% and 13.6%.

**Training from pre-trained model.** We provide the baseline results and our method when training from a ImageNet pre-trained model in Tab. 2. Despite having a strong feature extractor from the very beginning, it still needs to be adapted to discriminate better between fine-grained classes. We report results for 5, 10, and 20 equal tasks. *AdaGauss* achieves state-of-the-art results. It improves the average accuracy of the second-best method EFC [24] by 4.8% and 4.4% points on CUB200 and StanfordCars for $T = 10$, respectively. The results are consistent for other number of tasks.

**Ablation study.** We perform ablation of our method on CIFAR100 and ImagenetSubset datasets split into ten equal tasks in Table 3. First, we test our method with the nearest mean classifier (NMC) instead of the Bayes classifier to verify whether considering covariance improves the results. Without covariance matrices and with NMC [33] (1st row), we get worse results: 9.7% and 9.6% points lower average accuracy on CIFAR100 and ImagenetSubset, respectively. Memorizing covariances and sampling pseudo-prototypes to adapt means (2nd row) improves NMC results only slightly. Next, we utilize the Bayes classifier instead of NMC but assume that class distributions have diagonal covariance matrices (3rd row). That decreases the average accuracy of our method by 5.0% and 3.9%, respectively, proving that ground truth test distributions have non-zero off-diagonal. Then, we test our method without adapting means (5th row) like in IL2A [51] method. That severely hurts the performance - average accuracy decreases by 21.5% and 27.2 %. On the contrary, if we adapt means but not covariances like in EFC [24], we lose far less, 3.2% and 3.1%, respectively. Lastly, we check the performance of our method without the $L_{AC}$ component. To allow covariance matrices to be invertible, we add a shrink value of 0.5, similarly to [10]. This results in an average accuracy drop of 5.9% and 4.0%. The results are also consistent with the average incremental accuracies. This ablation proves our design choices and that all components are necessary to get the best results.

Table 3: Ablation of *AdaGauss* indicating the contribution from the different components. * signifies that we utilized covariance matrix shrinking with the value of 0.5 (chosen on the validation set) instead of anti-collapse loss to overcome the covariance matrix singularity problem.

| Classifier | Cov. Matrix | Adapt mean | Adapt covariance | $L_{AC}$ | CIFAR-100 (T=10) | | ImagenetSubset (T=10) | |
|---|---|---|---|---|---|---|---|---|
| | | | | | $A_{last}$ | $A_{inc}$ | $A_{last}$ | $A_{inc}$ |
| NMC | None | ✓ | ✓ | ✓ | 36.4 | 54.0 | 41.5 | 57.9 |
| NMC | Full | ✓ | ✓ | ✓ | 37.6 | 54.8 | 42.3 | 58.7 |
| Bayes | Diagonal | ✓ | ✓ | ✓ | 41.1 | 56.3 | 45.5 | 61.3 |
| Bayes | Full | ✗ | ✗ | ✗ | 22.9 | 42.8 | 22.5 | 43.4 |
| Bayes | Full | ✗ | ✓ | ✓ | 24.6 | 44.7 | 23.9 | 44.9 |
| Bayes | Full | ✓ | ✗ | ✓ | 42.9 | 57.7 | 48.0 | 62.7 |
| Bayes | Full | ✓ | ✓ | ✗* | 40.2 | 56.2 | 46.7 | 57.1 |
| Bayes | Full | ✓ | ✓ | ✓ | **46.1** | **60.2** | **51.1** | **65.0** |

## 4.2 Adaptation results

We verify how our adaptation method improves the quality of memorized class distributions on ImagenetSubset split into ten equal tasks. For this purpose, we measure the average distances between memorized and real classes after each task. More precisely, we measure the L2 distance between means and covariances as well as symmetrical Kulbach-Leibler divergence ($D_{KL}$) between memorized and real distributions. We utilize projected distillation ($\lambda = 10$) and compare our method to a baseline that does not adapt distributions like in [51, 52] (No adapt) and to the prototype drift compensation introduced in EFC [24] that adapts only means. We provide results in Fig. 5. We can see that our approach allows us to better approximate ground truth distributions. More precisely, compared to EFC, it decreases the distance to real-mean by ≈29%, to real-covariance by ≈39% and $D_{KL}$ distance by ≈72%. We can also see that the EFC approach does not improve distance to real-covariance compared to no adaptation, which is a drawback of this method.

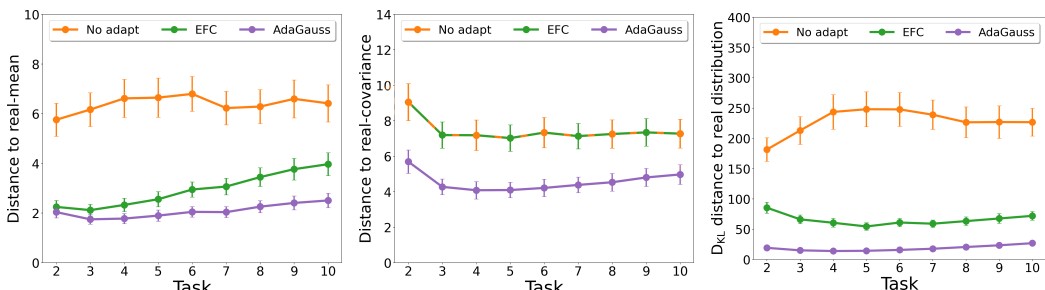

Figure 5: Distances from memorized distributions to the real ones in terms of distributions' mean, covariance and KL divergence across 10 tasks on ImagenetSubset dataset. AdaGauss greatly reduces errors and allows for better adaptation than prototype drift compensation (EFC).

## 4.3 Analysis of anti-collapse loss

We analyze the impact of anti-collapse $L_{AC}$ regularization term on ImagenetSubset split to 10 equal tasks. After the last task, we verify how much $L_{AC}$ improves the distribution of classes' covariance eigenvalues. We report results in Fig. 6. Without utilizing $L_{AC}$, the largest eigenvalue is $\approx 1.2 * 10^5$ times greater than the lowest, showcasing the dimensionality collapse. However, with $L_{AC}$, this difference equals ≈84, proving that more eigenvectors contribute towards representations, and the collapse is greatly diminished.

Next, we measure the average rank of covariance matrices memorized in each task for different knowledge distillation methods and projected distillation with $L_{AC}$. Here, we set $S = 64$. In Fig. 7 can see that without $L_{AC}$, all distillation methods present in existing methods struggle to achieve class covariance equal to latent size $S$, which according to Sec. 3.2 results in task-recency bias. Interestingly, when combining projected distillation with $L_{AC}$, the rank of covariance matrices equals 64 for each task, proving that $L_{AC}$ is a promising approach for combating dimensionality collapse when training from scratch.

An alternative method for overcoming singularity in covariance matrices is shrinking [10]. In Fig. 8, we present results for our method with different values of shrink performed when calculating covariance matrices on CIFAR100. Intuitively, increasing the shrink value decreases the method's efficacy, as it artificially alters the covariance to be different from the ground truth representation. Without using $L_{AC}$ and without shrink, it is impossible to invert the matrices, resulting in the crash of the method. Nevertheless, the results are the highest when utilizing $L_{AC}$ without shrink (60.2%).

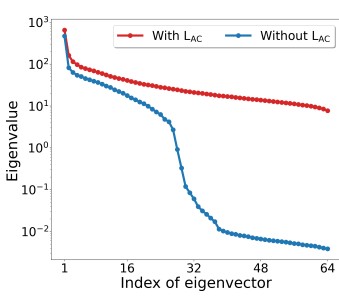 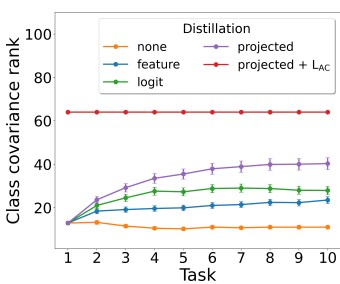 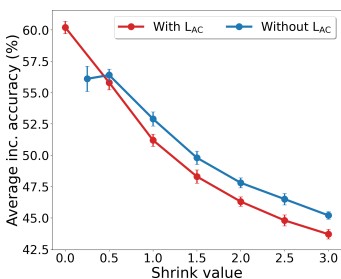

Figure 6: Distribution of eigenvalues of class representations for our method with and without $L_{AC}$ (anti-collapse loss) term. $L_{AC}$ greatly reduces the difference between the most and least significant eigenvalues, thus preventing dimensional collapse.

Figure 7: Ranks of classes' covariance matrices with different distillation methods and projected distillation with anti-collapse term (red) for latent space size $S = 64$. $L_{AC}$ makes covariance ranks to be equal to $S$ in every task.

Figure 8: Average incremental accuracy for *AdaGauss* for different values of covariance shrinking, with and without anti-collapse regularization. Results without $L_{AC}$ and shrink were not included due to the inability to invert covariance matrices after the first task.

## 4.4 Different distillation techniques

We test the performance of the projected distillation against other distillation techniques in *AdaGauss*. We train from scratch on CIFAR100, ImagenetSubset and utilize the pre-trained model on CUB200. We split datasets into ten equal tasks and use hyperparameters from experiments in Tab. 1 and Tab. 2. We present the results in Fig. 9. Projected distillation achieves better average accuracy than logit distillation by 1.4%, 0.9%, and 4.0% points on CIFAR100, ImagenetSubset, and CUB200, respectively. Interestingly, the gap between projected distillation and not using knowledge distillation is much lower on CUB200, which we contribute to using a strong pre-trained model.

## 4.5 Memory requirements

Our method does not increase the number of feature extractor's parameters. In addition, the adapter and distiller are discarded after the training, thus not increasing memory during the long CIL sessions and evaluations. $AdaGauss$ requires $S + \frac{S(S-1)}{2}$ parameters to memorize the mean and covariance of a class, where $S$ is the latent space size. Therefore, the method requires the same number of parameters as FeCAM [10] and fewer weights than EFC [24] as we do not expand the linear classifier. Additionally, $S$ can be decreased using linear bottleneck layer before the latent space.

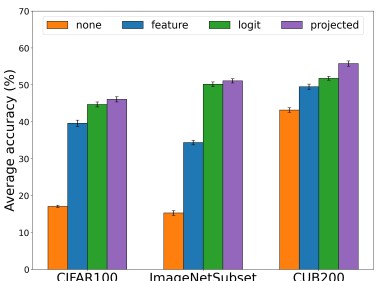

Figure 9: Average last task acc. of our method for different knowledge distillation techniques.

## 4.6 Time complexity of AdaGauss

We measure the training and inference time of popular EFCIL methods using their original implementations on a single machine with NVIDIA GeForce RTX 4060 and AMD Ryzen 5 5600X CPU. We repeat each experiment 5 times, train all methods for 200 epochs, use four workers, and have a batch size equal to 128. We test vanilla AdaGauss and AdaGauss, where the Bayes classifier is replaced with a trained linear head, where the classifier is trained on samples from class distributions (mean and cov. matrix). We utilize the FeTrIL version with a linear classification head.

We present results in Tab. 4. The inference of our method takes a similar amount of time as in FeCAM, as the feature extraction step is followed by performing Bayes classification. The inference

time of AdaGauss is slightly higher than that of methods with linear classification head (LwF, FeTrIL, AdaGauss with linear head) because Bayes classification requires an additional matrix multiplication when calculating the Mahalanobis distance.

The training time of AdaGauss is longer than for LwF, EFC, FeCAM, and FeTriL as we do not freeze the backbone after the initial task and additionally train the auxiliary adaptation network. Still, AdaGauss takes less time to train than its main competitor - EFC, and is much faster than SSRE. Our method does not increase the number of networks' parameters because the distiller and the adapter are disposed after training steps.

Table 4: Time complexity of EFCIL methods measured on CIFAR100 split into 10 tasks.

|  | LwF | FeTrIL | FeCAM | SSRE | EFC | AdaGauss | AdaGauss (lin. head) |
|---|---|---|---|---|---|---|---|
| Inference time (sec) | 66.3±2.7 | 68.3±3.4 | 74.2±4.0 | 274.7±12.3 | 67.4±3.1 | 72.8±3.2 | 65.3±2.2 |
| Training time (min) | 53.8±2.9 | 5.3±0.3 | 5.9±0.4 | 548.1±17.4 | 94.±2.9 | 86.3±3.2 | 103.3±4.2 |

## 5 Conclusions and limitations

In this work, we analyze the impact of dimensionality collapse in EFCIL. We explain that it leads to differences across tasks in ranks of classes' covariance matrices, which in turn causes task-recency bias. We also present that due to distribution drift, means and covariance of classes change, and they should be adapted from task to task. Based on these findings, we propose the first EFCIL method to adapt both means and covariances, dubbed *AdaGauss*. It utilizes feature distillation through a learnable projector and a novel anti-collapse regularization term during training that prevents having degenerated, non-invertible features covariance matrices as class representations. That, in turn, alleviates the task-recency bias of the classifier in continual learning. With the series of experiments, we show that *AdaGauss* achieves state-of-the-art results in common EFCIL scenarios, both when trained from scratch and when initialized from a pre-trained model.

The limitation of our method is that the cross-entropy separates classes only from the current task. However, when training the feature extractor, old classes can begin overlapping with each other and with new classes int he latent space causing forgetting. This problem is an open question in EFCIL. We speculate it can be alleviated wit a contrastive loss. Another problem arises when there is very little data representing a single class, making high-dimensional covariance matrix impossible to calculate. We tackle it by introducing a bottleneck layer at the very end of the feature extractor. However, it can limit its representational strength.

## Acknowledgments

The research of Grzegorz Rypeść was supported by the National Science Centre (Poland), PRE-LUDIUM 23 grant no. 2024/53/N/ST6/03018. This research was partially funded by National Science Centre, Poland, grant no: 2020/39/B/ST6/01511, 2022/45/B/ST6/02817, and 2023/51/D/ST6/02846. Bartłomiej Twardowski acknowledges the grant RYC2021-032765-I. This paper has been supported by the Horizon Europe Programme (HORIZON-CL4-2022-HUMAN-02) under the project "ELIAS: European Lighthouse of AI for Sustainability", GA no. 101120237. We gratefully acknowledge Polish high-performance computing infrastructure PLGrid (HPC Center: ACK Cyfronet AGH) for providing computer facilities and support within computational grant no. PLG/2023/017431.

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

# A  Additional experiments

## A.1  Adaptation results when starting from a pretrained model

We evaluate how our adaptation method improves the quality of memorized class distributions on CUB200 [42] split into ten equal tasks when starting from a model pre-trained on ImageNet. For this purpose, we measure the average distances between memorized and real classes after each task. More precisely, we measure the L2 distance between means and covariances as well as symmetrical Kulbach-Leibler divergence ($D_{KL}$) between memorized and real distributions. We utilize projected distillation ($\lambda = 10$) and compare our method to a baseline that does not adapt distributions like in [51, 52] (No adapt) and to the prototype drift compensation introduced in EFC [24] that adapts means only (EFC). We provide results in Fig. 10. Results are consistent with Fig. 10 - *AdaGauss* improves memorized distributions after every task.

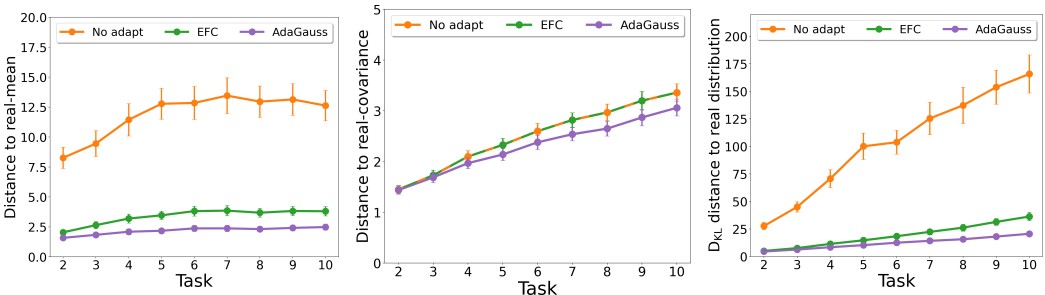

Figure 10: L2 distances from memorized distributions to the real ones in terms of distributions' mean, covariance and KL divergence across 10 tasks on CUB-200 dataset. The feature extractor was pre-trained on ImageNet.

## A.2  Impact of anti-collapse loss on optimization

Training the feature extractor of *AdaGauss* combines three loss functions: cross-entropy $L_{CE}$, knowledge distillation through a learnable projector $L_{PKD}$, and anti-collapse term to prevent features from dimensional collapse $L_{AC}$. We analyze average values of these losses during training of our method on ImagenetSubset (we set hyperparameters as in Tab. 1). Additionally, we modify Eq. 2 to incorporate strength of covariance regularization ($\beta$ hyperparameter):

$$L_{AC} = -\frac{1}{S}\sum_{i=1}^{S}\min(a_i, \beta) \tag{5}$$

In all of our experiments, we utilized $\beta = 1$ as this was sufficient to prevent dimensional collapse. Here, we test *AdaGauss* for $\beta = \{0.1, 1, 10, 100\}$.

We present results in Fig. 11. We can see that for $\beta = 1$, all losses are stable and consistently decrease. $L_{AC}$ decreases to -1.0, a value for which it is clipped. However, when increasing $\beta$, $L_{CE}$ and $L_{KD}$ become bigger, underfitting our approach. This results in much lower average and incremental accuracies. On the other hand, decreasing $\beta$ to 0.1 is not sufficient to prevent task-recency bias resulting in decreased accuracies.

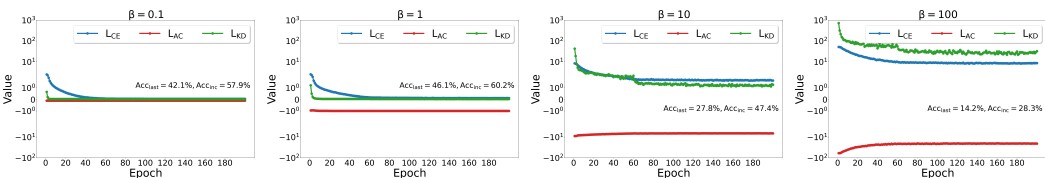

Figure 11: Value of $L_{CE}$, $L_{PKD}$, $L_{AC}$ losses for different $\beta$ parameters, last task average accuracy and average incremental accuracy.

## A.3 Tab. 1 and Tab. 2 with variance

We report the mean and variance of results reported in Tab. 1 when training from scratch in Tab. 5. Also, we report results from Tab. 2 when training from a pre-trained model in Tab. 6. Although we utilize additional anti-collapse loss compared to other methods, variance of accuracies achieved by *AdaGauss* is simillar to EFC.

Table 5: Average incremental and last accuracy in EFCIL scenarios when training the feature extractor from scratch. We report means and variances of 5 runs. We denote the best results **in bold**.

| Method | CIFAR-100 | | | | TinyImageNet | | | | ImagenetSubset | | | |
| | $T$=10 | | $T$=20 | | $T$=10 | | $T$=20 | | $T$=10 | | $T$=20 | |
| | $A_{last}$ | $A_{inc}$ | $A_{last}$ | $A_{inc}$ | $A_{last}$ | $A_{inc}$ | $A_{last}$ | $A_{inc}$ | $A_{last}$ | $A_{inc}$ | $A_{last}$ | $A_{inc}$ |
|---|---|---|---|---|---|---|---|---|---|---|---|---|
| EWC [17] | 31.2±2.9 | 49.1±1.3 | 17.4±2.4 | 31.0±1.2 | 17.6±1.5 | 32.6±1.2 | 11.3±1.2 | 26.8±1.1 | 24.6±4.1 | 39.4±3.1 | 12.8±2.0 | 27.0±1.0 |
| LwF [21] | 32.8±3.1 | 53.9±1.7 | 17.4±0.7 | 38.4±1.1 | 26.1±1.3 | 45.1±0.9 | 15.0±0.7 | 32.9±0.5 | 37.7±2.5 | 56.4±1.0 | 18.6±1.7 | 40.2±0.4 |
| PASS [52] | 30.5±1.0 | 47.9±1.9 | 17.4±0.7 | 32.9±1.0 | 24.1±0.5 | 39.3±0.9 | 18.7±1.4 | 32.0±1.8 | 26.4±1.3 | 45.7±0.2 | 14.4±1.2 | 31.7±0.4 |
| IL2A [51] | 31.7±1.3 | 48.4±2.0 | 23.0±0.9 | 40.2±1.1 | 25.3±0.9 | 42.0±1.7 | 19.8±1.8 | 35.5±2.3 | 27.7±1.8 | 48.4±1.5 | 17.5±1.6 | 34.9±0.7 |
| SSRE [53] | 30.4±0.7 | 47.3±1.9 | 17.5±0.8 | 32.5±1.1 | 22.9±1.0 | 38.8±2.0 | 17.3±1.1 | 30.6±2.0 | 25.4±1.2 | 43.8±1.1 | 16.3±1.1 | 31.2±1.5 |
| FeTrIL [31] | 34.9±0.5 | 51.2±1.1 | 23.3±0.8 | 38.5±1.1 | 31.0±0.9 | 45.6±1.7 | 25.7±0.6 | 39.5±1.2 | 36.2±1.2 | 52.6±0.6 | 26.6±1.5 | 42.4±2.1 |
| FeCAM [10] | 32.4±0.4 | 48.3±0.9 | 20.6±0.7 | 34.1±1.1 | 30.8±0.8 | 44.5±1.5 | 25.2±0.6 | 38.3±1.1 | 38.7±1.0 | 54.8±0.5 | 29.0±1.3 | 44.6±2.0 |
| EFC [24] | 43.6±0.7 | 58.6±0.9 | 32.2±1.3 | 47.3±1.4 | 34.1±0.8 | 48.0±0.6 | 28.7±0.4 | 42.1±1.0 | 47.4±1.4 | 59.9±1.4 | 35.8±1.7 | 49.9±2.1 |
| AdaGauss | **46.1±0.8** | **60.2±0.9** | **37.8±1.5** | **52.4±1.4** | **36.5±0.9** | **50.6±0.8** | **31.3±1.0** | **45.1±1.2** | **51.1±1.2** | **65.0±1.4** | **42.6±1.6** | **57.4±1.9** |

Table 6: Average incremental and last accuracy in EFCIL fine-grained scenarios when utilizing a pre-trained feature extractor on ImageNet. The means and variance of 5 runs are reported with the best results **in bold**.

| Method | CUB200 | | | | | | StanfordCars | | | | | |
| | $T$=5 | | $T$=10 | | $T$=20 | | $T$=5 | | $T$=10 | | $T$=20 | |
| | $A_{last}$ | $A_{inc}$ | $A_{last}$ | $A_{inc}$ | $A_{last}$ | $A_{inc}$ | $A_{last}$ | $A_{inc}$ | $A_{last}$ | $A_{inc}$ | $A_{last}$ | $A_{inc}$ |
|---|---|---|---|---|---|---|---|---|---|---|---|---|
| EWC [17] | 21.6±0.4 | 38.2±0.3 | 15.8±0.7 | 32.6±0.5 | 12.3±0.8 | 27.2±0.6 | 24.3±0.6 | 44.0±0.6 | 14.3±0.8 | 34.5±0.7 | 10.9±1.2 | 27.9±1.1 |
| LwF [21] | 44.3±0.7 | 57.7±0.7 | 30.4±1.1 | 46.1±1.0 | 19.4±1.6 | 34.7±1.8 | 39.0±0.8 | 55.2±0.6 | 28.0±1.0 | 46.5±1.0 | 14.7±1.5 | 30.5±1.4 |
| PASS [52] | 34.5±0.5 | 48.6±0.4 | 27.0±0.9 | 42.3±0.9 | 18.1±1.2 | 36.9±1.1 | 33.3±1.0 | 48.9±0.9 | 26.4±1.1 | 41.0±0.8 | 13.9±1.6 | 28.1±1.2 |
| IL2A [51] | 36.9±1.2 | 51.3±1.4 | 29.4±1.3 | 45.5±1.7 | 20.8±2.0 | 35.1±2.3 | 39.4±1.3 | 49.1±1.5 | 27.3±1.6 | 45.1±1.7 | 14.2±2.3 | 28.7±2.8 |
| FeTrIL [31] | 41.9±0.5 | 53.2±0.6 | 36.9±0.7 | 48.2±0.6 | 34.6±1.0 | 45.3±0.9 | 46.0±0.7 | 58.5±1.0 | 40.5±0.8 | 53.4±0.9 | 32.5±1.1 | 43.3±1.3 |
| FeCAM [10] | 43.5±0.5 | 56.0±0.7 | 40.2±0.8 | 54.9±1.0 | 36.2±1.1 | 48.9±1.3 | 45.3±0.5 | 58.0±0.6 | 41.4±0.8 | 55.2±0.9 | 34.0±1.1 | 46.0±1.2 |
| EFC [24] | 58.3±0.4 | 68.9±0.6 | 51.0±0.6 | 63.3±0.7 | 46.1±1.0 | 59.3±1.3 | 50.1±0.5 | 63.2±0.7 | 43.1±0.8 | 57.6±0.7 | 28.1±1.8 | 48.2±1.6 |
| AdaGauss | **60.4±0.5** | **69.2±0.7** | **55.8±0.5** | **66.2±0.8** | **47.4±0.8** | **60.6±1.0** | **53.3±0.7** | **64.0±1.0** | **47.5±0.9** | **58.5±1.1** | **34.8±1.2** | **48.6±1.3** |

## A.4 Different architecture of pretrained backbone

We test AdaGauss with different feautre extractors, namely ViT small and ConvNext. The results, presented in Tab. 7, are for EFCIL setting with 10 and 20 equal tasks and weights pretrained on ImageNet (as in Tab. 2). Using more modern feature extractors architectures further improves the results of AdaGauss.

Table 7: *AdaGauss* results with different backbone architectures. We report last accuracy | average accuracy.

| | CUB200 | | FGVCAircraft | |
| | T=10 | T=20 | T=10 | T=20 |
|---|---|---|---|---|
| Resnet18 | 55.8 \| 66.2 | 47.4 \| 60.6 | 47.5 \| 58.5 | 34.8 \| 48.6 |
| ConvNext (small) | 63.4 \| 73.1 | 47.9 \| 64.1 | 49.3 \| 62.9 | 37.3 \| 51.4 |
| ViT (small) | 68.2 \| 77.5 | 48.9 \| 67.5 | 48.0 \| 60.6 | 35.7 \| 50.2 |

## A.5 Half dataset results

Learning from scratch is more challenging than half-dataset setting as it requires to incrementally train feature extractor, not just the classifier. However, using the pre-trained model (or learning from half) can be considered a more practical and real-life setting. Thus, we evaluated our method with a pre-trained model in Table 2. However, we have additionally compared our method to the mentioned baselines in a half-dataset setting using the original implementations under the same data augmentations as AdaGauss. Please note that we did not have enough time to perform hyperparameter search for our method - we utilized these from the equal task setting, whereas the results for other methods were optimized by their authors. We provide results in the Tab. 8.

AdaGauss performs better than PASS, SSRE, and FeTrIL (5 tasks) in the half-dataset setting. However, it is slightly worse than most recent baselines when using default hyperparameters. FeCAM, ACIL, and DS-AL freeze the feature extractor after the initial task, which can explain their good results in the half-dataset training.

Table 8: Half dataset in the first task results. We report last accuracy | average accuracy.

|  | CIFAR100 | | | | ImageNetSubset | | | |
| --- | --- | --- | --- | --- | --- | --- | --- | --- |
|  | T = 5 | | T = 10 | | T = 5 | | T = 10 | |
| PASS | 54.5 | 61.8 | 53.8 | 60.9 | 57.9 | 64.4 | 58.2 | 61.8 |
| SSRE | 55.7 | 63.9 | 54.9 | 63.2 | 58.3 | 65.2 | 61.4 | 67.7 |
| FeTrIL | 58.3 | 65.1 | 56.2 | 64.6 | 65.6 | 72.8 | 65.3 | 72.1 |
| FeCAM | 60.2 | 67.2 | 59.9 | 66.9 | 67.3 | 75.3 | 67.6 | 74.9 |
| ACIL | 57.8 | 66.3 | 57.7 | 66.0 | 67.0 | 74.8 | 67.2 | 74.6 |
| DS-AL | 61.4 | 68.4 | 61.4 | 68.4 | 68.0 | 75.2 | 67.7 | 75.1 |
| AdaGauss | 58.9 | 65.7 | 55.4 | 63.7 | 66.8 | 74.1 | 62.8 | 68.0 |

## A.6 Predicted semantic shift for classes

Here, we verify whether our adaptation network can predict different shift for different classes in experiments from Tab.1. We test on CIFAR100 for $T = 10$ and the answer is positive, as shown in Fig. 12.

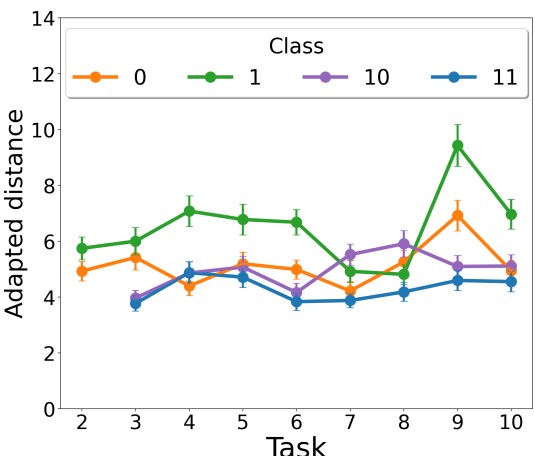

Figure 12: Predicted shift for different classes on CIFAR100 (10 equal tasks) by *AdaGauss*. The Euclidean distance is measured between old and new position in each task.

## A.7 Batch norm influence on AdaGauss

We measure the accuracy achieved by AdaGauss in the EFCIL scenario from scratch (CIFAR100, ImageNetSubset) and pretrained (CUB200). We train for 10 tasks without batch norm layers and with frozen batch norm layers. Results are provided in Tab. 9. We can see that possesing batch-norm layers in Resnet18 is beneficial.

Table 9: *AdaGauss* results without or with frozen batch-norm. We report last accuracy and average accuracy separated by |.

|  | CIFAR100 | ImageNetSubset | CUB200 |
| --- | --- | --- | --- |
| Resnet18 (no BN) | 44.6±0.7 \| 58.1±0.7 | 49.1±0.8 \| 61.7±0.9 | 54.2±0.5 \| 66.1±0.7 |
| Resnet18 (freezed BN) | 45.3±0.7 \| 58.7±0.9 | 49.4±0.8 \| 62.0±1.0 | 55.2±0.4 \| 65.7±0.6 |
| Resnet18 | 46.1±0.8 \| 60.2±0.9 | 51.1±1.0 \| 65.0±1.2 | 55.8±0.5 \| 66.2±0.8 |

