# OpenReview forum: "Task-recency bias strikes back: Adapting covariances in Exemplar-Free Class Incremental Learning"
_NeurIPS.cc/2024/Conference — NeurIPS 2024 poster_

### Official Review · Reviewer_VMJn · 2024-07-05

**Soundness:** 2
**Presentation:** 3
**Contribution:** 2
**Rating:** 5
**Confidence:** 4

**Summary:**

This paper addresses the Exemplar-Free Class Incremental Learning (EFCIL) challenge. It identifies two critical issues that undermine the effectiveness of existing methodologies and proposes a novel approach, AdaGauss, which adapts covariance matrices from task to task and mitigates task-recency bias.

**Strengths:**

1. The paper is generally well-written and well motivated. It clearly demonstrates that the changes of covariance matrices also matters in Exemplar-Free Class Incremental Learning.
2. The proposed method demonstrates pretty good experimental results on all five datasets whether training from scratch or starting from a pre-trained backbone.
3. The proposed method is straightforward and easy to understand.

**Weaknesses:**

1. I appreciate the efforts the authors have devoted to detailing the three observations encountered in Exemplar-Free Class Incremental Learning. However, observations 2 and 3 appear quite similar to me, as both seem to represent a simplification of the representation for previously seen classes. Additionally, the analysis of observations 2 and 3 does not significantly depart from existing explanations for why classification results tend to skew towards recent tasks, a phenomenon (task-recency bias / representation forgetting) already well-documented in the broader field of continual learning.
2. The proposed idea that both the mean and covariance should be adapted during training shares similarities with test-time adaptation methods. Therefore, some comparisons are necessary to delineate these relationships further.
3. Although it is quite straightforward that encouraging the feature extractor to produce features with linearly independent dimensions can mitigate dimensionality collapse, this approach does not guarantee the production of meaningful features. Additionally, it remains unclear whether simply optimizing the covariance matrices of features from one mini-batch can ensure linear independence. Some theoretical analysis would better explain the effectiveness of the proposed loss term $L_{AC}$ in mitigating dimensionality collapse (which I think there do exist some possible results to derive).

**Questions:**

See the weakness.

---

> ### Author Rebuttal · Authors · 2024-08-06
>
> We appreciate the feedback provided by the Reviewer. We will now address the specific weaknesses indicated:
>
> **W1: Novelty of explanation of the task-recency bias**
>
> We agree that task recency bias was well explored in CIL literature. However, in most works [1-4], the focus is on the bias in the linear classifier, which CIL methods try to unskew. On the contrary, in our work, we explain the bias in the embedding space - before the linear classifier and a novel method to prevent it ($L_{AC}$). Observations 2 and 3 show that ranks of covariance matrices increase in the later tasks, leading inverses of covariance matrices to have higher norms. That, in fact, generates task-recency bias in methods that utilize Bayes classifier or sample from class distribution in the embedding space. To the best of our knowledge, we are first to explain this and to provide a simple solution in the form of $L_{AC}$ loss. Less elegant solutions involve techniques like shrinking covariance matrices (used in FeCAM and EFC) as they introduce more hyperparameters.
>
>
> **W2: Comparison to TTA methods**
>
> Some TTA approaches, e.g., CAFA [5] or TTAC [6] align the means and covariances at test-time, with those computed during pretraining. However, there exist notable differences compared to the EFCIL setting considered in our work. In TTA, data of all the classes are available during the adaptation phase (it is only the features that change, like in domain incremental learning), whereas in EFCIL, the model is learning new classes in new tasks, and therefore, a class-level feature alignment cannot be applied (as there is no access to previous task classes). However, it is true that there are some similarities (e.g., the feature drift that occurs in TTA due to changes in domains), and we will add references and discussion to TTA works in our work.
>
> **W3.1: Impact of $L_{AC}$ on meaningfullness of features**
>
> Our loss function consists of three components, which are of a different optimization goal, and a sweet spot between them must be found. As shown in Eq. 5 and Fig.11 in the Appendix, we analyzed the $\beta$ hyperparameter, which sets the strength for the covariance regularization. As the Reviewer noticed, increasing the $\beta$ hyper-parameter increases the value of the cross-entropy loss, meaning that features produced by the network are less meaningful (from the classification perspective). Therefore, a good $\beta$ must be found during hyperparameter search. In all of our experiments, we utilized $\beta=1$. We have made the tradeoff between loss functions clearer in the new revision.
>
> **W3.2: Explaining linear independence of features**
>
> We verified the linear dependency problem experimentally. The value of  $L_{AC}$ loss remains at -1 after each epoch in our experiments, as presented in the Appendix, Fig.11 ($\beta$=1). That means that for each minibatch the values of the diagonal of its Cholesky decomposition are greater than 1, and thus are positive. The latter implies that the covariance matrix of each minibatch is full rank [7]. Thus, the features are linearly independent. We agree that optimizing $L_{AC}$ at the mini-batch level is a stochastic problem like SGD, and it also requires that each mini-batch should be representative of the train dataset distribution. However, it results in the overall full-rank covariance matrices (what is presented in Figure 6 and 7 in the main paper). In our work, we use a mini-batch size that is four times bigger than the latent dimension to perform Cholesky decomposition efficiently.
>
> ----
> We hope our response alleviates any concerns the Reviewer may have. However, if there are any remaining uncertainties, kindly specify references and additional questions, and we can further discuss them. Otherwise, we would appreciate if the Reviewer reconsidered improving the final score of our submission.
>
> ----
> [1] Wu, Yue, et al. "Large scale incremental learning." Proceedings of the IEEE/CVF conference on computer vision and pattern recognition. 2019.
> [2] Zhou, Da-Wei, et al. "Deep class-incremental learning: A survey." arXiv preprint arXiv:2302.03648 (2023).
> [3] Castro, Francisco M., et al. "End-to-end incremental learning." Proceedings of the European conference on computer vision (ECCV). 2018.
> [4] Hou, Saihui, et al. "Learning a unified classifier incrementally via rebalancing." Proceedings of the IEEE/CVF conference on computer vision and pattern recognition. 2019.
> [5] Su, Yongyi, Xun Xu, and Kui Jia. "Revisiting realistic test-time training: Sequential inference and adaptation by anchored clustering." NeurIPS 2022.
> [6] Jung, Sanghun, et al. "Cafa: Class-aware feature alignment for test-time adaptation." ICCV 2023.
> [7] Horn, Roger A.; Johnson, Charles R. (1985). Matrix Analysis. Cambridge University Press. ISBN 0-521-38632-2

---

> > ### Comment · Reviewer_VMJn · 2024-08-09
> > **Response**
> >
> > I appreciate the authors taking the time to respond and I tend to keep my score.

---

> ### Author Response · Authors · 2024-08-12
>
> Thank you for your response! We think we have addressed most of your concerns. We are eager to assist if there's anything else we can do to improve your score. Reviewer xJqZ has changed the score based on the valuable discussion!

---

### Official Review · Reviewer_6bmT · 2024-07-09

**Soundness:** 3
**Presentation:** 3
**Contribution:** 3
**Rating:** 6
**Confidence:** 5

**Summary:**

Existing methods use Gaussian distributions to represent classes in the feature extractor's latent space, but face unchanged covariance matrices and task-recency bias. This paper introduces AdaGauss, an approach that adapts covariance matrices and mitigates the bias through an anti-collapse loss function. AdaGauss achieves top performance on EFCIL benchmarks and datasets, whether training from scratch or using a pre-trained backbone.

**Strengths:**

1. The paper presents innovative approaches to tackle the EFCIL problem.
2. The writing is quite good, and easy to follow overall.
3. he paper includes comprehensive experimental findings to support its claims.

**Weaknesses:**

1. Literature is incomplete. The paper concentrates on the EFCIL, yet quite a number of important EFCIL techniques [1-3] are not presented, and I want to see experimental comparisons with [1,3] if possible.

[1]Huiping Zhuang, Zhenyu Weng, Hongxin Wei, Renchunzi Xie, Kar-Ann Toh, Zhiping Lin”ACIL: Analytic Class-Incremental Learning with Absolute Memorization and Privacy Protection”, Thirty-Sixth Conference on Neural Information Processing Systems (NeurIPS) 2022.

[2] Ma, C.; Ji, Z.; Huang, Z.; Shen, Y.; Gao,M.; and Xu, J. 2023. Progressive Voronoi Diagram Subdivision Enables Accurate Data-free Class-Incremental Learning. In The Eleventh International Conference on Learning Representations.

[3] Zhuang H, He R, Tong K, et al. DS-AL: A dual-stream analytic learning for exemplar-free class-incremental learning[C]//Proceedings of the AAAI Conference on Artificial Intelligence. 2024, 38(15): 17237-17244.

2. The claim "dimensionality collapse" concept is very important, but the authors didn't explain anything related.

3. An adaptor is needed for implementing the proposed algorithm. Could you provide a comparison among the proposed methods regarding the auxillary networks imposed?

**Questions:**

see weakness.

**Limitations:**

n.a

---

> ### Author Rebuttal · Authors · 2024-08-06
>
> We thank the Reviewer for providing important references, constructive feedback, and insightful comments. Below we respond to the weaknesses mentioned.
>
> **W1: Incomplete literature and comparison to ACIL and DS-AL**
>
> Thank you for providing relevant literature - we have added it to the Related Works of the paper. Thanks to the good quality of the code in [1, 3] we have compared our method to ACIL and DS-AL. Please note that [1, 2, 3] methods utilize half-dataset warm-start settings in the original papers. Firstly, we ran [1, 3] in the **equal** task setting using original implementations. The results ($A_{last}$ | $A_{inc}$) are below:
>
>
> |  | CIFAR100  | CIFAR100 | ImageNetSubset | ImageNetSubset |
> | -------- | :--------: | :--------: | :--------: | :--------: |
> |  | T = 10 | T = 20 | T = 10 | T = 20 |
> | ACIL[1]     | 38.8 \| 53.2    | 30.8 \| 42.7     | 44.2 \| 54.8    | 35.3 \| 47.6 |
> | DS-AL[3]     | 40.8 \| 54.9 | 31.7 \| 43.2     | 46.8 \| 58.6   | 36.7 \| 48.5 |
> | AdaGauss     | **46.1** \| **60.2** | **37.8** \| **52.4** | **51.1** \| **65.0** | **42.6** \| **57.4** |
>
> Next, we have run AdaGauss with the half dataset in the first task setting:
>
> |  | CIFAR100  | CIFAR100 | ImageNetSubset | ImageNetSubset |
> | -------- | :--------: | :--------: | :--------: | :--------: |
> |  | T = 5 | T = 10 | T = 5 | T = 10 |
> | ACIL[1]     | 57.8 \| 66.3     | 57.7 \| 66.0   | 67.0 \| 74.8    | 67.2 \| 74.6 |
> | DS-AL[3]     | **61.4** \| **68.4**  | **61.4** \| **68.4**     | **68.0** \| **75.2**   | **67.7** \| **75.1** |
> | AdaGauss     | 58.9 \| 65.7     | 55.4 \| 63.7     | 66.8 \| 74.1    | 62.8 \| 68.0|
>
> We can see that *AdaGauss* performs better in the equal task scenario. However, in the half-dataset scenario, ACIL and DS-AL are better. In our opinion, the half-dataset setting prefers methods with a frozen feature extractor (FE) after the first task [1, 2, 3]. However, in this paper, we focus on explaining the task recency bias when training the FE in incremental steps. We also did not have enough time to perform a hyperparameter search. We have added these results and the discussion to the Appendix.
>
> **W2: Explaining the dimensionality collapse**
> We tried to explain this concept in line 73: *a large fraction of features' variance is described only by a small fraction of their dimensions.* This is also reflected by the analysis done in Section 3.2, line 109, where we presented that the rank of the class covariance matrix is far lower than its dimensionality. That is the result of training the feature extractor with cross-entropy that forces the representational collapse of the latent representation to the number of classes that need to be linearly separable [5]. However, we agree that this can still be improved. We have done that in the introduction in our work and included more related work.
>
> **W3: Different adaptor architectures**
> We have included results ($A_{last}$ | $A_{inc}$) for different types of adaptors in the appendix and below. $d$ denotes number of times the hidden layer is bigger than the input and output layers. We train for 10 equal tasks.
>
> |  | CIFAR100 | ImageNetSubset |
> | :--------: | :--------: | :--------: |
> | SDC[4]     | 43.7 $\pm$ 0.6     | 46.7 $\pm$ 0.8     |
> | Linear     | 42.3 $\pm$ 0.6    | 45.5 $\pm$ 0.7    |
> | MLP (2-layers, d=4)     | 45.7 $\pm$ 0.8     | 50.4 $\pm$ 0.8     |
> | MLP (2-layers, d=16)     | **46.1 $\pm$ 0.8**     | **51.1 $\pm$ 1.1**     |
> | MLP (3-layers, d=4)     | 44.6 $\pm$ 0.7     | 49.9 $\pm$ 1.0     |
> | MLP (3-layers, d=16)     | 45.1 $\pm$ 0.8     | 50.1 $\pm$ 0.8     |
>
> We can see that utilizing non-linear MLP networks yields better results than linear network and SDC methods. 2-layer MLP networks are also preferred over the 3-layer ones.
>
> ---
> If we have adequately addressed the Reviewer's concerns, we kindly ask for your support and slightly improving the score. If you have any further concerns or additional points to raise, we are eager to address them. Your insights are valuable in enhancing the quality and impact of our research.
>
> ---
> [1] Huiping Zhuang, Zhenyu Weng, Hongxin Wei, Renchunzi Xie, Kar-Ann Toh, Zhiping Lin ”ACIL: Analytic Class-Incremental Learning with Absolute Memorization and Privacy Protection”, Thirty-Sixth Conference on Neural Information Processing Systems (NeurIPS) 2022.
> [2] Ma, C.; Ji, Z.; Huang, Z.; Shen, Y.; Gao,M.; and Xu, J. 2023. Progressive Voronoi Diagram Subdivision Enables Accurate Data-free Class-Incremental Learning. In The Eleventh International Conference on Learning Representations.
> [3] Zhuang H, He R, Tong K, et al. DS-AL: A dual-stream analytic learning for exemplar-free class-incremental learning[C]//Proceedings of the AAAI Conference on Artificial Intelligence. 2024, 38(15): 17237-17244.
> [4] Yu, Lu, et al. "Semantic drift compensation for class-incremental learning." Proceedings of the IEEE/CVF conference on computer vision and pattern recognition. 2020.
> [5] Vardan Papyan, XY Han, and David L Donoho. Prevalence of neural collapse during the terminal phase of deep learning training. Proceedings of the National Academy of Sciences, 2020.

---

> > ### Comment · Reviewer_6bmT · 2024-08-09
> > **Thank you for the response**
> >
> > Thank you for the response. The rebuttal has addressed my concerns and I will keep my rating.

---

> ### Author Response · Authors · 2024-08-12
>
> We are glad to hear that our response addressed all of your concerns! We are here to assist if there's anything else we can do to improve your score. Please note, that Reviewer xJqZ has just modified the score based on the valuable discussion!

---

### Official Review · Reviewer_5j6M · 2024-07-12

**Soundness:** 3
**Presentation:** 3
**Contribution:** 3
**Rating:** 6
**Confidence:** 5

**Summary:**

The paper addresses the problem of Exemplar-Free Class Incremental Learning (EFCIL), which involves training a model on sequential tasks without access to past data. Current methods represent classes as Gaussian distributions in the feature space, enabling Bayes classification or pseudo-feature replay for classifier training. However, these methods face issues such as the need to adapt covariance matrices after each task and susceptibility to task-recency bias due to dimensionality collapse. The authors propose AdaGauss, a novel method that adapts covariance matrices between tasks and incorporates an anti-collapse loss function to mitigate task-recency bias. AdaGauss achieves state-of-the-art performance on popular EFCIL benchmarks and datasets, whether training from scratch or using a pre-trained backbone.

**Strengths:**

Overall this is a good paper with the following strengths:
* This paper provides a clear experimental analysis and elaboration of the motivation for the proposed method.
* For EFCIL, it is important and interesting to track changes in the distribution of past classes.
* The proposed method achieves good results in multiple settings.

**Weaknesses:**

* The method of knowledge distillation through projectors proposed in this paper has been widely studied and discussed [1, 2]. Different from the general field of knowledge distillation, what are the technical innovations of the proposed method? Does it have unique insights for continual learning tasks?
* The proposed method introduces several additional structures and requires a large number of samples from the simulated distribution to be trained, so the number of model parameters and the required training time should be discussed.

[1] Chen Y, Wang S, Liu J, et al. Improved feature distillation via projector ensemble. NeurIPS 2022.

[2] Miles R, Mikolajczyk K. Understanding the role of the projector in knowledge distillation. AAAI 2024.

**Questions:**

I have some questions mainly about the experiments:

1. The experiments in this paper use the "learn from scratch" setting. Another common setting used in CIL is the "learn from half", which is also a dominant setting in EFCIL. It simply means that half of the classes in the dataset are learned as an initial task, and then the remaining classes are evenly divided into subsequent incremental learning tasks. This experimental setting is used in most of the compared works, such as PASS, SSRE, FeTrIL and FeCAM. How does AdaGauss perform in this setting?
2. Although the authors provide a detailed ablation analysis in Table 3, I still have some concerns. Why is $L_{AC}$  replaced by "utilized covariance matrix shrinking with the value of 0.5" in the sixth row of table 3, instead of just removing it? Are there other components of the proposed method that cannot be run in the absence of $L_{AC}$?
3. Does the adaptive tuning for Gaussian distributions fail if there is no $L_{AC}$ to resist dimensionality collapse?
4. Additionally, it would be beneficial if the authors could include a row comparing the performance of AdaGauss without both "Adapt mean", "Adapt covariance" and $L_{AC}$. This additional row would contribute to a clearer comparison and enhance the transparency of the evaluation.

**Limitations:**

It is hard to foresee any potential negative societal impact of this theoretical work.

---

> ### Author Rebuttal · Authors · 2024-08-06
>
> We express our gratitude to the Reviewer for the feedback and insightful remarks. We shall begin by addressing the specific weaknesses pointed out:
>
> **W1: Innovation of knowledge distillation through a projector**
>
> In this work, we focus on knowledge distillation in EFCIL, which is different than [1, 2]. KD with the projector for EFCIL is only one part of the proposed solution. Together with adaptation step and anti-collapse loss the method receives the best results (Tab. 3).
>
> Additionally, our unique insight about KD is explaining how it impacts the task-recency bias in EFCIL. As presented in Section 3.2, KD through a learnable projector is less susceptible to bias because it provides stronger representations. That, combined with $L_{AC}$, allows us to overcome the bias and achieve good results. We have made that clearer in the new version of the manuscript.
>
> **W2: Number of parameters and time complexity**
>
> The number of model parameters is discussed in Section 4.2: *Memory requirements*. After the training, the projectors for KD and adaptation are removed and not stored. Therefore, the number of network parameters is not increased.
>
> We aggregated logs from our experiments on CIFAR100 split into 10 equal tasks to compare the training and inference time. We utilized original implementations of baseline methods and utilized Resnet18 trained for 200 epochs with batch size of 128 on a single machine. We used NVIDIA GeForce RTX 4060 and AMD Ryzen 5 5600X CPU for the experiments; we repeated them 5 times. We use a linear head classifier for FeTrIL and also test a version of *AdaGauss* with a linear head classifier instead of a Bayes classifier - we sample from memorized distributions to train it. We present results in Tab.2 of the pdf file.
>
> Methods that freeze the backbone after the first task (FeTrIL and FeCAM) have lower training time than others. AdaGauss takes less time to train than EFC, as it does not require training the linear head.
>
> The inference time of methods that utilize the Bayes classifier (FeCAM, AdaGauss) is higher than methods that utilize linear classification heads (LwF, FeTrIL, EFC). Replacing the Bayes classifier with a linear head in *AdaGauss* boosts its inference time by 7.5 seconds.
>
> We have added these results and discussion to the Appendix.
>
> **Q1: Learn from half dataset**
>
> Learning from scratch is more challenging than half-dataset setting as it requires to incrementally train feature extractor [3], not just the classifier. However, using the pre-trained model (or learning from half) can be considered a more practical and real-life setting. Thus, we evaluated our method with a pre-trained model in **Table 2**. However, we have additionally compared our method to the mentioned baselines in a half-dataset setting using the original implementations under the same data augmentations as *AdaGauss*. Please note that we did not have enough time to perform hyperparameter search for our method - we utilized these from the equal task setting, whereas the results for other methods were optimized by their authors. We provide results in the form ($A_{last}$ | $A_{inc}$) in the Tab. 4 of the rebuttal pdf file.
>
> *AdaGauss* performs better than PASS, SSRE, and FeTrIL (5 tasks) in the half-dataset setting. However, it is slightly worse than most recent baselines when using default hyperparameters. FeCAM, ACIL, and DS-AL freeze the feature extractor after the initial task, which can explain their good results in the half-dataset training. We have added these results to the Appendix.
>
> **Q2.1: Why is $L_{AC}$ replaced by shrinking in Tab. 3?**
>
> Suppose we just remove the $L_{AC}$. In that case, covariance matrices of classes become singular, and it is mathematically impossible to invert them, leading to the inability to perform Bayes classification or to sample from these distributions (required to adapt distributions). Therefore, we utilized the lowest possible shrink value to invert covariance matrices (in this case 0.5; we performed a grid search to obtain it from values 0.01, 0.05, 0.1, 0.5, ...). Shrinkage is a standard technique used in FeCAM[4] and EFC[3] to alleviate the problem of singularity and matrix inversion.
>
> **Q2.2: Are there other components of the method that cannot be run in the absence of $L_{AC}$**
>
> Without $L_{AC}$ we cannot perform classification and adaptaing Gaussians. Both of these require the inverse of class covariance matrices, and they cannot be calculated when covariance matrices are singular. $L_{AC}$ prevents that. We have improved the writing to emphasize this important aspect of the AdaGauss method and made it more clear in the ablation study.
>
> **Q3: Does the adaptive tuning for Gaussian distributions fail if there is no $L_{AC}$**
> Yes, without $L_{AC}$, class covariance matrices are singular, and we cannot invert them. Therefore, we cannot sample from class distributions (step 16. in Alg. 1).
>
>
> **Q4: Providing additional row to the ablation study**
> We have added this row and the discussion to the new version of the Appendix. We also present the results ($A_{last}$ | $A_{inc}$) in Tab.5 of the pdf file. With all of these components turned off, *AdaGauss* achieves poor results compared to the baseline.
>
> ----
>
> If the Reviewer's concerns have been sufficiently addressed in our responses, we humbly seek their support for the paper and improving the score.
>
>  ----
>
> [1] Chen Y, Wang S, Liu J, et al. Improved feature distillation via projector ensemble. NeurIPS 2022.
> [2] Miles R, Mikolajczyk K. Understanding the role of the projector in knowledge distillation. AAAI 2024.
> [3] Elastic Feature Consolidation For Cold Start Exemplar-Free Incremental Learning. ICLR 2024
> [4] Fecam: Exploiting the heterogeneity of class distributions in exemplar-free continual learning. NeurIPS 2024

---

> ### Comment · Reviewer_5j6M · 2024-08-09
> **Response**
>
> I appreciate the authors' response. This rebuttal addresses most of my concerns. Although AdaGauss does not achieve SOTA performance in the "Learn from half" setting, solving EFCIL with an adaptable Gaussian distribution sounds rational. I think adjusting the loss weight of knowledge distillation might be helpful to improve performance in the "Learn from half" setting. Taking into account the rebuttal and other reviewers' comments, I tend to keep my rating.

---

> ### Author Response · Authors · 2024-08-12
>
> Thank you for your response! We are glad that we have addressed most of your concerns. If there is any issue we can resolve to improve your score, we are eager to do it. Please note that the Reviewer xJqZ changed the score based on the discussion!

---

### Official Review · Reviewer_xJqZ · 2024-07-13

**Soundness:** 3
**Presentation:** 2
**Contribution:** 3
**Rating:** 6
**Confidence:** 4

**Summary:**

This paper analyzes the impact of dimensionality collapse in EFCIL and examines the distribution shift of the mean and covariance matrix. Based on these findings, the paper proposes the AdaGauss method, which adapts the covariance and mean, and designs a loss term to prevent the dimensionality collapse of the feature extractor.

**Strengths:**

- The analysis of the dimensionality collapse of the feature extractor is interesting.
- The paper provides a comprehensive experimental analysis, which clearly demonstrates the effectiveness of different components within the proposed model.

**Weaknesses:**

- The writing in the current version needs improvement. There are many parts that I had to reread several times to understand. For example, the three observations in Lines 89-118, I strongly recommend to summarize them in one sentence initially to highlight the core findings. In Section 3.3.2, which is one of the most important parts and contributions of the paper, the explanation is insufficient even though I refer to the appendix. What is the meaning of $a_i$ ? There is no clear definition.
- Regarding Figure 1, is it merely an illustrative figure, or is it a visualization figure? If it is a visualization figure, what are the details used to depict this figure?
- In Section 3.3.4, the learned adaptation network is trained on the t-th task. Why can it be used for previous μ and σ? The underlying assumption is that all previous tasks share the same shift pattern. Is this a reasonable assumption?
- For Algorithm 1, I am not sure how it addresses the Batch Norm statistics, which are crucial to continual learning and the covariance shift this paper focuses on [1,2].
- The time complexity of the proposed method compared to other methods should be discussed.


[1] Continual Normalization: Rethinking Batch Normalization for Online Continual Learning, ICLR 2022.
[2] Diagnosing batch normalization in class incremental learning, 2022.

**Questions:**

1. Please address the questions in the weakness.
2. Can the proposed method be applied to other network structures such as 4-layer convolutional networks or ViT?

---

> ### Author Rebuttal · Authors · 2024-08-06
>
> We thank the Reviewer for the insightful comments and providing relevant works. We begin by responding to the weaknesses:
>
> **W1: Writing improvements**
> We have revisited the whole Section 3 and applied the Reviewer's suggestions in the new version of the manuscript.
>
> **W1: Meaning of $a_{i}$**
> The definition of $a_i$ is provided in line 146: *More precisely, let S be the size of the feature vectors and $a_i$ be the i-th element of the diagonal of a Cholesky decomposition of minibatch’s covariance matrix.* We have made it clearer in the new version.
>
> **W2: Regarding Fig. 1**
> Fig. 1 is an illustration that provides intuition into the problem of mean and covariance adaptation; it is not a visualization. However, the provided accuracy after the last task and KL divergence (between memorized and real class distributions) were measured in experiments on the ImageNetSubset dataset split into ten tasks. Such illustrations are utilized in a variety of works, e.g. [1] (Fig. 1), [2] (Fig. 1), [3] (Fig. 3), [4] (Fig. 1c).
>
> **W3: Assumption that all tasks share the same shift pattern**
> In EFCIL, we do not have access to old data. Thus, we learn the non-linear adaptation network with current task data and use its generalization for the old classes' distribution (μ and $\Sigma$). We do not assume that the shift pattern for all the previous tasks is the same. We train an additional **non-linear** network to learn these patterns, assuming that we have only access to the current data. That is the main assumption. The learned adapter network estimates the shift **differently** for each task and its classes, as we visualize in Fig. 1 of the rebuttal file (classes 0, 1 are from 1st task, 10 and 11 - 2nd). On the contrary, in the recent EFC [4], only the mean is adapted, and there is an assumption that the covariance does not shift (Fig. 5).
>
> **W4: Regarding batch norm**
> We agree that BN is an important aspect of CL. However, it is more severe for methods that do not perform shift compensation, which is reflected in the provided works [5, 6]. The fact that BN changes the shift of class representations is fine in *AdaGauss*, as we utilize the MLP adaptor to predict and compensate for this shift. Therefore, our method is agnostic to BN, and BN does not negatively impact *AdaGauss*. To prove it, we provide (Tab.1 in rebuttal pdf) results ($A_{last}$ | $A_{inc}$) with frozen after the first task or removed BN layers from Resnet18. We can see that removing or freezing BN layers slightly decreases the performance of the method.
> Additionally, our method is architecture agnostic and can work with networks in which the BN is absent, i.e., ViT, which uses Layer Norm. We provide results with different backbones below in the response to Question 2. We have added a discussion about BN and mentioned RW [5, 6] to our manuscript.
>
> **W5: Time complexity**
> We have gathered cumulative inference and training time of our experiments on CIFAR100 split into ten tasks. We utilize the original implementation for each method and run experiments on a single machine with NVIDIA GeForce RTX 4060 and AMD Ryzen 5 5600X CPU. We repeat each experiment 5 times, train all methods for 200 epochs, use 4 workers and batch size equal to 128. We test vanilla *AdaGauss*, and *AdaGauss* where the Bayes classifier is replaced with a trained linear head, where the classifier is trained on samples from class distributions (mean and cov. matrix). We utilize FeTrIL version with linear classification head.
>
> We present results in Tab.2 of the rebuttal pdf file. The inference of our method takes a similar amount of time as in FeCAM, as the feature extraction step is followed by performing Bayes classification. The inference time of *AdaGauss* is slightly higher than that of methods with linear classification head (LwF, FeTrIL, *AdaGauss* with linear head) because Bayes classification requires an additional matrix multiplication when calculating the Mahalanobis distance.
>
> The training time of *AdaGauss* is longer than for LwF, EFC, FeCAM, and FeTriL as we do not freeze the backbone after the initial task and additionally train the auxiliary adaptation network. Still, *AdaGauss* takes less time to train than its main competitor - EFC, and is much faster than SSRE. Our method does not increase the number of networks' parameters because the distiller and the adapter are disposed after training steps.
>
> **Q2: Can the proposed method be applied to other network structures?**
> Yes, the method is architecture-agnostic and can be applied to different network structures and different training regimes. We do not assume any requirements towards backbone architecture. We will make it clearer in the paper, and we will include AdaGauss results for different backbones in the appendix, namely ViT small and ConvNext. These results ($A_{last}$ | $A_{inc}$), presented in Tab.3 of the pdf file, are for EFCIL setting with 10 and 20 equal tasks and weights pretrained on ImageNet (as in Tab. 2). Using more modern feature extractors architectures further improves the results of AdaGauss.
>
> ---
> We hope that our explanation has addressed the Reviewer's concerns. Should there be any additional queries, we are willing to provide further details. If no further clarification is needed, we kindly ask the Reviewer to increase the final score.
>
> ---
> [1] Fecam: Exploiting the heterogeneity of class distributions in exemplar-free continual learning. NeurIPS 2024.
> [2] Fetril: Feature translation for exemplar-free class-incremental learning. WACV 2023.
> [3] Semantic drift compensation for class-incremental learning. CVPR 2020.
> [4] Elastic Feature Consolidation For Cold Start Exemplar-Free Incremental Learning. ICLR 2024
> [5] Continual Normalization: Rethinking Batch Normalization for Online Continual Learning, ICLR 2022.
> [6] Diagnosing batch normalization in class incremental learning, 2022.

---

> > ### Comment · Reviewer_xJqZ · 2024-08-11
> > **Comment by Reviewer xJqZ**
> >
> > I thank the authors for the efforts to address the concerns, especially the additional experiments conducted to verify the effectiveness of the proposed method with different backbone networks. The clarifications made by the authors and the results of the additional experiments addressed my concerns. I have increased my score to 6 accordingly.

---

> ### Author Response · Authors · 2024-08-12
>
> Thank you for improving the score. Your review helped us improve the work a lot!

---

### Author Rebuttal · Authors · 2024-08-06

We want to express our gratitude to all the Reviewers and Chairs for their dedication and effort. The majority of Reviewers consider accepting the work, all agree on comprehensive experimental analysis, and three Reviewers highlight the motivation behind the method (5j6M, VMJn, 6bmT). We have thoroughly reviewed the feedback and addressed all the raised concerns, what significantly enhanced the quality of our submission.

We have now prepared a revised version of our work and are ready to engage in further discussions. Based on the comments received, we have performed additional experiments, which are presented in the rebuttal pdf file, and made the following changes to the revised version:

* Improved introduction and method section to better explain the problem of task recency bias caused by dimensionality collapse in EFCIL, and why it is different than popular recency bias in the classification head (xJqZ, 6bmT, VMJn).
* Added analysis of training and inference time complexity  of *AdaGauss* compared to other methods (xJqZ, 5j6M).
* Tested our method in a half-dataset setting and added results to the Appendix (5j6M, 6bmT).
* Added discussion of relevant works proposed by reviewers (6bmT, xJqZ).
* To confirm that *AdaGauss* works with different backbone architectures, we have run experiments with ConvNext and ViT and added them to the Appendix (xJqZ).
* Experimentally proved that the predicted distribution shift is different for different classes (xJqZ).
* To understand the impact of batch norm (BN) layers on our method we have performed experiments with frozen and removed BN layers. We have added a discussion to the Appendix (xJqZ).
* Provided *AdaGauss* results for different feature adaptation networks (6bmT).

We thank the Reviewers once again and look forward to discussing any other aspects of the paper that require further clarification.
-- authors

---

> ### Author Response · Authors · 2024-08-12
>
> Once more, we thank the reviewers and AC for their hard work. Based on the responses, we have clarified all doubts and weaknesses raised by the Reviewers. Our work contributes to Continual Learning as it is **first** to explain the task recency bias on the embedding level, **first** to introduce the anti-collapse loss to prevent it, and **first** to adapt class covariances from task to task. There is still time for the rebuttal, and we are willing to participate in further discussion.

---

### Comment · Area_Chair_rday · 2024-08-11

Dear Reviewers,

The deadline of reviewer-authors discussion is approaching. If you have not done so already, please check the rebuttal and provide your response at your earliest convenience.

Best wishes,

AC

---

### Decision · Program_Chairs · 2024-09-25

**Decision:**

Accept (poster)

**Comment:**

After discussion, all reviewers lean towards acceptance for this work with two borderline scores. The reviewers recognized that the problem is significant, the proposed method to be straightforward and effective with strong experimental results. However, the reviewers still have some remaining concerns, including no new insights for the community and lacking some baselines that do not freeze the feature extractor. In my view, I tend to recommend acceptance for this work and the authors should add those baselines in the final version.